# Potential of Secondary Metabolites of *Diaporthe* Species Associated with Terrestrial and Marine Origins

**DOI:** 10.3390/jof9040453

**Published:** 2023-04-07

**Authors:** Wei Wei, Babar Khan, Qun Dai, Jie Lin, Liqin Kang, Nasir Ahmed Rajput, Wei Yan, Guiyou Liu

**Affiliations:** 1School of Life Sciences and Chemical Engineering, Jiangsu Second Normal University, Nanjing 211200, China; 2College of Plant Protection, State & Local Joint Engineering Research Center of Green Pesticide Invention and Application, Nanjing Agricultural University, Nanjing 210095, China; 3Department of Plant Pathology, University of Agriculture, Faisalabad 38000, Pakistan

**Keywords:** *Diaporthe*, secondary metabolites, biological potencies, drug discovery

## Abstract

*Diaporthe* species produce versatile secondary metabolites (SMs), including terpenoids, fatty acids, polyketides, steroids, and alkaloids. These structurally diverse SMs exhibit a wide range of biological activities, including cytotoxic, antifungal, antibacterial, antiviral, antioxidant, anti-inflammatory, and phytotoxic activities, which could be exploited in the medical, agricultural, and other modern industries. This review comprehensively covers the production and biological potencies of isolated natural products from the genus *Diaporthe* associated with terrestrial and marine origins. A total of 275 SMs have been summarized from terrestrial (153; 55%) and marine (110; 41%) origins during the last twelve years, and 12 (4%) compounds are common to both environments. All secondary metabolites are categorized predominantly on the basis of their bioactivities (cytotoxic, antibacterial, antifungal, and miscellaneous activity). Overall, 134 bioactive compounds were isolated from terrestrial (92; 55%) and marine (42; 34%) origins, but about half the compounds did not report any kind of activity. The antiSMASH results suggested that *Diaporthe* strains are capable of encoding a wide range of SMs and have tremendous biosynthetic potential for new SMs. This study will be useful for future research on drug discovery from terrestrial and marine natural products.

## 1. Introduction

*Diaporthe* is an important fungal genus of plant pathogens [1] belonging to the family Diaporthaceae, order Diaporthales, and class Sordariomycetes [2]. It is isolated mainly from plant hosts, which are distributed worldwide; many of them have been reported as plant pathogens, nonpathogenic endophytes, or saprobes, and human and other mammalian pathogens [3,4]. *Diaporthe* sp. is a widespread fungal genus that colonizes a wide range of hosts. It consists of nearly 800 described species, with around 950 species being attributed to its asexual state (*Phomopsis*) [5]. It is often isolated from above-ground plants, especially tropical and temperate woody plants [6]. Among numerous endophytic fungi, the genus *Diaporthe* is known for its potent biosynthetic ability to produce bioactive metabolites [7,8]. Secondary metabolites (SMs) isolated from *Diaporthe* sp. have shown a wide range of biological activities and chemical structures [9,10]. Chemical studies on some *Diaporthe* spp. have revealed a variety of bioactive natural products [11], such as cytotoxic diapolic acids [12], antifungal compounds [5,13], antibacterial agents [14,15], anti-candidal ketone derivatives [16], and anti-tubercular metabolites [17]. In the last twelve years, a total of 106 bioactive SMs have been reported from the genus *Diaporthe* [18].

Endophytic communities that develop inside the host plants are influenced by various parameters, such as environmental conditions (terrestrial and marine), host type, etc. [19]. Fungal endophytes are asymptomatic inhabitants of plant tissues that have the capability to colonize all parts of plants and determine their functional aspects, including increasing plant growth, acting as a biocontrol agent, naturally protecting the host from pests, and enduring tolerance against numerous biotic/abiotic stresses [20,21]. In return, they benefit from host plants in several ways, including providing nutrients, protection from desiccation, spatial structure, and passing on reproductive fungal propagules to the next generation of hosts in the case of vertical transmission [22]. Due to the vast diversity of endophytic fungal communities, the characterization of the SMs of each endophytic fungal community is difficult; therefore, the current review aims to describe the SMs species from the genus *Diaporthe* from two main origins (terrestrial and marine) and, furthermore, to classify them on the basis of their biological potency.

## 2. Terrestrial Origin

### 2.1. Cytotoxic Metabolites

Liu et al. (2013) isolated nine compounds (**1**–**9**), including a novel (1*R*,2*E*,4*S*,5*R*)-1-[(2*R*)-5-oxotetrahydrofuran-2-yl]-4,5-dihydroxy-hex-2-en-1-yl(2*E*)-2 methylbut-2-enoate (**1**), a known (*1R,2R,4R*)-trihydroxy-*p*-menthane (**2**), three linear furanopolyketides (**3**–**5**), and four lovastatin analogues, oblongolides D (**6**), H (**7**), P (**8**), and V (**9**), from *Diaporthe* sp. SXZ-19 on *C. acuminate*. These compounds showed weak cytotoxic activities against HCT 116 cells at a concentration of 10 µM [23]. Two bioactive metabolites, emodin (**10**) and arbutin (**11**), were isolated from an endophytic fungus *D. lithocarpus*. Compound **10** exhibited remarkable cytotoxic activity against P-388 murine leukemia cells (IC_50_ = 0.41 µg/mL), and **11** showed moderate cytotoxicity against murine leukemia P-388 cells and had an IC_50_ value at 2.91 µg/mL [15]. Two cytoskyrin-type bisanthraquinones, cytoskyrin C (**12**), and (+)-epicytoskyrin (**13**), were isolated from *Diaporthe* sp., an endophytic fungus derived from *Anoectochilus roxburghii*. Both compounds showed dose-dependent cytotoxicities against SMMC-7721 cells [24]. A new compound, vochysiamide B (**14**), and the known 2,5-dihydroxybenzyl alcohol (**15**) were derived from *D. vochysiae* LGMF1583 on the medicinal plant *Vochysia divergens* and showed cytotoxic activities against A549 human non-small cell lung and PC3 human prostate cell lines [8]. Mycoepoxydiene (**16**) and eremofortin F (**17**) were obtained from the endophytic fungus *Diaporthe* sp. SNB-GSS10 on *Sabicea cinerea* and showed cytotoxic activity against KB and MRC5 cells [6].

Two eremophilanes, lithocarins B (**18**) and C (**19**), were isolated from an endophytic fungus *D. lithocarpus* A740 on *Morinda officinalis*. Both compounds exhibited cytotoxicity against SF-268, MCF-7, HepG-2, and A549 tumor cells with IC_50_ values between 37.68 and 97.71 µM [9]. The endophytic fungus *D*. *terebinthifolii* GG3F6, derived from the medicinal plant *Glycyrrhiza glabra*, was a source of the metabolite xylarolide (**20**), which showed cytotoxicity against MIAPaCa-2, HCT-116, and T47D cancer cells with IC_50_ values of 38 µM, 100 µM, and 7 μM, respectively [12]. The metabolites xylarolide A (**21**) and xylarolide (**20**) were isolated from the fungus *Diaporthe* sp. on *D*. *inoxia* and showed remarkable cytotoxicity against MIAPaCa-2 with IC_50_ values of 20 μM and 32 μM, respectively, and against PC-3 with IC_50_ values of 14 μM and 18 μM, respectively [25]. Brissow et al. (2017) obtained 18-des-hydroxy cytochalasin H (**22**) from the endophytic fungus *D. phaseolorum*-92C on *Combretum lanceolatum*. This compound exhibited cytotoxic activity against the breast cancer cells MDA-MB-231 and MCF-7 [26]. A new brasilane-type sesquiterpenoid, diaporol R (**23**), was isolated from an endophytic *Diaporthe* sp. on leaves of *R. stylosa*. Diaporol R had a moderate cytotoxic effect on SW480 cancer cells and exhibited an IC_50_ value of 8.72 ± 1.32 µM [27]. Diaporone A (**24**), a new dihydroisocoumarin derivative, was isolated from the crude extract of the plant endophytic fungus *Diaporthe* sp. and exhibited weak cytotoxicity against the human cervical cancer (HeLa) cell line with an IC_50_ value of 97.4 μM [28]. Yang et al. (2020) isolated nine cytochalasans (**25**–**33**) from the endophytic fungus *Diaporthe* sp. SC-J0138 isolated from the leaves of *Cyclosorus parasiticus*. All compounds showed cytotoxic activity [29]. Khan et al. (2023) isolated a novel compound phomopthane A (**34**) from the plant-derived fungus *D. unshiuensis* YSP3, which exhibited cytotoxic activities against HeLa and MCF-7 cells with IC_50_ values of 5.92 µM and 7.50 µM, respectively [30]. Compounds **1**–**34** are shown in Figure 1.

### 2.2. Antibacterial Metabolites

Two isocoumarin metabolites, (10*S*)-diaporthin (**35**) and orthosporin (**36**), were isolated from *D. terebinthifolii* LGMF907 isolated from *Schinus terebinthifolius*. They showed antibacterial activities against methicillin-sensitive *Staphylococcus aureus* and methicillin-resistant *S. aureus* [31]. A new 3-substituted-5-diazenylcyclopentendione, named kongiidiazadione (**37**), was separated from *D. kongii* on *C. lanatus* and showed antibacterial activity against *Bacillus amyloliquefaciens* [32]. The three metabolites emodin (**10**), coumarin (**38**), and 1,2,8-trihydroxyanthraquinone (**39**) were isolated from the endophytic fungus *D. lithocarpus*. Compound **38** had a diameter inhibition zone of 12.3 ± 0.3 mm against the bacterium *B. subtilis*, and **10** showed antibacterial activity against *B. subtilis*, *M. luteus*, *Pseudomonas fluorescences*, *E. coli*, and *S. cerevisiae* with inhibition zone diameters of 14.7 mm, 13.2 mm, 13.7 mm, 12.7 mm, and 11.7 mm, respectively, while compound **39** displayed antibacterial activity against *B. subtilis*, *E. coli*, and *S. cerevisiae* with inhibition zone diameters of 14.2 mm, 11.3 mm, and 10.7 mm, respectively [15]. Two antibacterial metabolites, phomosines A (**40**) and C (**41**), were extracted from *Diaporthe* sp. F2934 of the plant *Siparuna gesnerioides*. Both were active against *S. aureus*, *M. luteus*, *Streptococcus oralis*, *Enterococcus fecalis*, *Enterococcus cloacae*, and *Bordetella bronchiseptica*, with the diameter of the zone of inhibition ranging from 6 ± 0.62 to 12 ± 1.18 mm at a concentration of 4 µg/µL [11].

A new lanostanoid, 19-nor-lanosta-5(10),6,8,24-tetraene- 1*α*,3*β*,12*β*,22*S*-tetraol (**42**), along with two known steroids, 3*b*,5*a*,9*a*-trihydroxy-(22*E*,24*R*)-ergosta-7,22-dien-6-one (**43**) and chaxine C (**44**), were isolated from *Diaporthe* sp. LG23 on the Chinese medicinal plant *Mahonia fortune*. Compound **42** exhibited antibacterial activity against both Gram-positive and Gram-negative bacteria, and **43** and **44** showed antibacterial activity against *B. subtilis* with streptomycin as a positive control [14]. Two new fatty acids, diapolic acids A and B (**45** and **46**), along with two known compounds, xylarolide (**20**) and phomolide G (**47**), were isolated from the endophytic fungus *D*. *terebinthifolii* GG3F6, which was derived from the medicinal plant *Glycyrrhiza glabra*. All these compounds show antibacterial activity against *Y. enterocolitica* with IC_50_ values of 78.4 μM, 73.4 μM, 72.1 μM, and 69.2 μM, respectively [12]. The new 21-acetoxycytochalasins J_3_ (**48**) was extracted from *Diaporthe* sp. GDG-118 on *Sophora tonkinensis* and showed moderate antibacterial activity against *Bacillus anthraci* and *Escherichia coli* [33]. A carboxamide, vochysiamide B (**14**), from *D*. *vochysiae* LGMF1583 showed antibacterial activity on the Gram-negative bacterium *Klebsiella pneumoniae* (KPC) with a minimum inhibitory concentration (MIC) value of 80 μg/mL [8]. Flavomannin-6,60-di-O-methyl ether (**49**) was extracted from an endophytic strain of *D. melonis* from *Annona squamosal*, which showed antimicrobial activity against *S. aureus* 25697, *S. aureus* 29213, and *Streptococcus pneumonia* ATCC 49619 with MIC values of 32 μg/mL, 32 μg/mL, and 2 µg/mL, respectively [34]. A phenolicmetabolite, tyrosol (**50**), was extracted from *D. helianthi* isolated from *Luehea divaricate*. Tyrosol showed significant antagonistic activity against several tested pathogenic bacterial strains [35]. Compound **24** was isolated from the plant endophytic fungus *Diaporthe* sp. and showed moderate antibacterial activity against *Bacillus subtilis* with a MIC value of 66.7 μM [28]. The novel 3-methoxy-5-methylnaphthalene-1, 7-diol (**51**) was isolated from a *Diaporthe* sp. on the plant *Syzygium cordatum*. Compound **51** demonstrated antibacterial activity against *Pseudomonas syringae pv phaseolicola* and *Xanthomonas axonopodis* pv *phaseoli*, with MIC values of 2.50 mg/mL (7.00 ± 0.00 mm) and 1.25 mg/mL (7.67 ± 0.33 mm), respectively, against test organisms [36]. A new alternariol methyl ether-12-O-α-D-arabinoside (**52**) derived from *D. unshiuensis* YSP3 and showed antibacterial effect on *B. subtilis* (MIC value 16 μg/mL) [30]. The structures of compounds **35**–**52** are shown in Figure 2.

### 2.3. Antifungal Secondary Metabolites

Tanney et al. (2016) isolated four secondary metabolites of *D. maritima* from healthy *Picea mariana* and *Picea rubens* needles, including phomopsolides A (**53**), B (**54**), and C (**55**) and a stable *a*-pyrone, (*S*,*E*)-6-(4-hydroxy-3-oxopent-1-en-1-yl)-2H-pyran-2-one (**56**). All compounds showed antifungal activities against *M. violaceum* and *Saccharomyces cerevisiae* [5]. A known product, 7-hydroxy-6-metoxycoumarin (**57**), was isolated from the endophytic fungus *D. lithocarpus*, showing significant antifungal activity against *Sporobolomyces salminocolor* with an inhibition zone of 12.2 ± 0.3 mm [15]. A *bis*-anthraquinone derivative, (+)-2,20-*epi*-cytoskyrin A (**58**), was isolated from *Diaporthe* sp. GNBP-10 from *Uncaria gambir* Roxb. It showed antifungal activity against 22 yeast strains and 3 filamentous fungi with MICs ranging from 16 μg/mL to 128 µg/mL [37]. Cytochalasins were isolated from *Diaporthe* sp. GDG-118, including 7-acetoxycytochalasin H (**59**) and cytochalasins H (**60**) and E (**61**), and showed varying degrees of antifungal activity against *Alternaria oleracea*, *Pestalotiopsis theae*, *Colletotrichum capsici*, and *Ceratocystis paradoxa* [33]. The novel metabolite 3-hydroxy-5-methoxyhex-5-ene-2,4-dione (**62**) was isolated from *Diaporthe* sp. ED2 on the herb *Orthosiphon stamieus* Benth. It showed antifungal activity against *C. albicans* with an MIC value of 3.1 μg/mL [16]. A new metabolite, eucalyptacid A (**63**), along with the three known metabolites cytosporone C (**64**), 1-(4-hydroxyphenyl) ethane-1,2-diol (**65**), and (2-hydroxy-2-phenylethyl) acetamide (**66**), was isolated from the solid rice cultures of the endophytic fungus *D. eucalyptorum* KY-9 that had been isolated from *Melia azedarach*. All compounds exhibited antifungal activities against *Alternaria solani* [13]. Compounds **53**–**66** are shown in Figure 3.

### 2.4. Miscellaneous Activities

Seven metabolites, mucorisocoumarin A (**67**); pestalotiopsone B (**68**); acetoxydothiorelone B (**69**); dothiorelones B (**70**), L (**71**), and G (**72**); and cytosporone D (**73**), were isolated from the endophytic fungus *D. pseudomangiferaea* on *Tylophora ouata*. Compounds **67**–**73** displayed anti-fibrosis activity with inhibition rates of 17.4%, 59.2%, 62.9%, 41.1%, 32.9%, and 52.1% in human lung fibroblast MRC-5 cell activation induced by TFG-b at 10 µM. Cytosporone D (**73**) showed antioxidant activity with an inhibition rate of 63.3% by releasing MOA at a concentration of 10 µM and moderate antidiabetic activity toward protein tyrosine phosphatase 1B (PTP1B) [38]. The fungus *D. eres* derived from pathogen-infected leaves of *Hedera helix* produced an isocoumarin, 3,4-dihydro-8-hydroxy-3,5-dimethylisocoumarin (**74**), and tyrosol (**50**), which had a phytotoxic effect on the growth of *Lemna paucicostata* [39]. A novel metabolite, diportharine A (**75**), was obtained from the culture of a *Diaporthe* sp. isolated from *Datura inoxia*. It showed remarkable antioxidant activity by scavenging DPPH radicals (EC_50_ = 10.3 µM) [25]. Two new benzopyranones, diaportheones A (**76**) and B (**77**), were extracted from *Diaporthe* sp. P133 from *Pandanus amaryllifolius*. They exhibited moderate antitubercular activities and achieved MIC values of 100.9 μM and 3.5 µM, respectively, against *Mycobacterium tuberculosis* H37Rv with rifampin as the positive control (MIC = 0.25 µM) [40]. The cyclohexeneoxidedione derivatives phyllostine acetate (**78**) and phyllostine (**79**) were extracted from *D. miriciae* on the plant *Cyperus iria* and showed potent antifeedant activities on *Plutella xylostella*. [41]. Cytoskyrin C (**12**) and (+)-epicytoskyrin (**13**) were isolated from *Diaporthe* sp. and were able to activate the NF-_K_B pathway and increase the relative activity of luciferase at a concentration of 50 µM [24]. Five phytotoxic compounds, p-cresol (**80**), 4-hydroxybenzoic acid (**81**), 4-hydroxybenzaldehyde (**82**), nectriapyrone (**83**), and tyrosol (**50**), were isolated from *D. eres* on *V. vinifera* wood. In leaf disk and leaf absorption bioassays, the phytotoxicities of all compounds increased with concentration over the range 0.1–1 mg/mL [42]. Two diphenyl ether derivatives, diaporthols A (**84**) and B (**85**), were extracted from *Diaporthe* sp. ECN-137 isolated from the leaves of *Phellodendron amurense*. Compounds displayed a migration inhibitory effect on TGF-*β*1-triggered MDA-MB-231 breast cancer cells at a concentration of 20 µM [43]. Two new metabolites, gulypyrone A (**86**) and phomentrioloxin B (**87**), were extracted from a strain of *D*. *gulyae* isolated from *C*. *lanatus*, which had a low phytotoxic effect and caused some necrosis in various weed and crop species [44]. Phomolide C (**88**) from a *Diaporthe* sp. on *Aucuba japonica* var*. borealis* inhibited the proliferation of human colon adenocarcinoma cells at a concentration of 50 μg/mL [45]. Compound 18-des-hydroxy cytochalasin H (**22**) from the endophytic fungus *D. phaseolorum*-92C inhibited leishmanicidal activity and moderate antioxidant activity against the breast cancer cells MDA-MB-231 and MCF-7 [26]. Studies of the strain *Diaporthe* sp. JC-J7 from the stems of *Dendrobium nobile* led to the isolation of a new compound, diaporthsin E (**89**). It showed low antihyperlipidemic activity on triglycerides (TG) in steatotic L-02 cells with an inhibition rate of 26% at a concentration of 5 μg/mL [46]. Two dibenzopyrones, 2-hydroxy-alternariol (**90**) and alternariol (**91**), were isolated from the endophytic fungus *Diaporthe* sp. CB10100. Both compounds significantly reduced the production of NO to as low as 10 μM in LPS-induced RAW264.7 cells [47]. A new metabolite, phomentrioloxin (**92**), was isolated from the liquid culture of *Phomopsis* sp. (asexual state of *Diaphorte*), which showed phytotoxic activity, and caused growth and chlorophyll content reduction in fronds of *Lemna minor* and inhibition of tomato rootlet elongation [48]. Structures of compounds **67**–**92** are shown in Figure 4.

### 2.5. Compounds with No Activity

Two known compounds (**93**–**94**) isolated from *D. lithocarpus* showed no activity [15]. The compound vochysiamides A (**95**) from *D. vochysiae* LGMF1583 did not report activity [8]. The endophytic fungus *D. pseudomangiferae* yielded the inactive compound altiloxin A (**96**) [6]. A new benzophenone derivative, named tenllone I (**97**), the new lithocarin D (**98**), and the known phomopene (**99**) were isolated from the endophytic fungus *D. lithocarpus* A740. These compounds were not found to be significantly active [9]. Xylarolide B (**100**) isolated from the culture of an endophytic fungus *Diaporthe* sp. Harbored from *Datura inoxia* showed no activity [25]. Nine new sesquiterpenoids, diaporols J–Q and S (**101**–**108** and **109**), were isolated from *Diaporthe* sp., an endophytic fungus. None of them reported any activity [27]. Alternariol 4,10-dimethyl ether (**110**) and alternariol 4-methyl ether (**111**) were isolated from a crude extract of the plant endophytic fungus *Diaporthe* sp. and did not display any kind of bioactivity [28]. Three compounds, 4H-1-benzopyra-4-one-2,3-dihydro-5-hydroxy-2,8-dimetyl (**112**), 4H-1-benzopyran-4-one-2,3-dihydro-5-hydroxy-8-(hydroxy-lmethyl)-2-methyl (**113**), and phomosine D (**114**), were isolated from the *Diaporthe* sp. F2934. These isolated compounds were found to be inactive [11]. Four known compounds, 3*β*,5*α*,9*α*,14*α*-tetrahydroxy-(22*E*,24*R*)-ergosta-7,22-dien 6-one (**115**), (22*E*,24*R*)-ergosta-7,9(11),22-triene-3*β*,5*α*,6*α*-triol (**116**), demethylincisterol A3 (**117**), and volemolide (**118**), were isolated from an endophytic fungus, *Diaporthe* sp. LG23, and were found to have no bioactivity [14]. A chemical investigation into the endophyte *D. melonis* reported the isolation of two new compounds, diaporthemins A (**119**) and B (**120**). Neither compound was reported to have any kind of potency [34]. Three inactive metabolites, a new metabolite, eucalactam B (**121**), and two known metabolites, eugenitol (**122**) and 4-hydroxyphenethyl alcohol (**123**), were isolated from the solid rice cultures of the endophytic fungus *D. eucalyptorum* KY-9 [13]. The chemical exploration of an endophytic fungus *D. pseudomangiferaea* led to the isolation of eleven inactive (**124**–**134**) secondary metabolites [38]. Nine compounds (**135**–**143**) were isolated from a strain of *D*. *gulyae*, but did not report any bioactivity [44]. Ten inactive polyketones (**144**–**153**) were isolated from the fermentation of *Diaporthe* sp. JC-J7 [46]. Nine inactive metabolites (**154**–**162**) were isolated from the endophytic fungus *Diaporthe* sp. CB10100 [47]. An inactive new cytochalasan (**163**) was isolated from the endophytic fungus *Diaporthe* sp. SC-J0138 [29]. Two inactive novel compounds, phomopthane B (**164**) and phomopyrone B (**165**), were isolated from *D. unshiuensis* [30]. The structures of compounds **93**–**165** are shown in Figure 5.

## 3. Marine Origin

### 3.1. Antibacterial and Antifungal Metabolites

A chemical investigation into *Diaporthe amygdali* SgKB4, an endophytic fungal strain isolated from the West Sumatran mangrove plant *Sonneratiagriffithii* Kurz, led to the isolation of cytochalasin H (**60**). This compound showed mild antibacterial activity against some pathogenic bacteria [49]. The fungus *D. phaseolorum* derived from *Laguncularia racemose*, afforded 3-hydroxypropionic acid (**166**), which showed antimicrobial activity against *S. aureus* and *S. typhi* [50]. A new compound (**167**), named diaporthelactone, was isolated from the culture of *Diaporthe* sp., a marine fungus growing in the submerged decayed leaves of *Kandelia candel* in the mangrove, and exhibited inhibitory antifungal activity against *Aspergillus niger* with a MIC of 50 µg/mL [51]. Niaz et al. (2021) isolated a new isochromophilone G (**168**) along with six known azaphilones (**169**–**174**) from the endophytic fungus *Diaporthe perseae* on the Chinese mangrove *Pongamia pinnata* (L.). All compounds exhibited antibacterial potency against human pathogens [52]. Compounds **166**–**174** are shown in Figure 6.

### 3.2. Miscellaneous Activities

Three compounds, pestalotiopsones F (**175**) and B (**176**), and 3,8-dihydroxy-6-methyl-9-oxo-9Hxanthene- 1-carboxylate (**177**), were isolated from *Diaporthe* sp. SCSIO 41011. These compounds showed significant anti-IAV activities against three influenza A virus subtypes, including A/Puerto Rico/8/34 H274Y (H1N1), A/FM-1/1/47 (H1N1), and A/Aichi/2/68 (H3N2) [53]. Phomoxanthone A (**178**), with a novel carbon skeleton, was isolated from the fungus *D. phaseolorum* FS431 and showed good cytotoxic potency against MCF-7, HepG-2, and A549 with IC_50_ values of 2.60 μM, 2.55 μM, and 4.64 µM, respectively [54]. A new compound biatriosporin N (**179**), together with five known compounds (**180**–**182, 60,** and **178**), was obtained from the culture of the fungus *Diaporthe* sp. GZU-1021. All compounds displayed significant inhibitory effects against NO production with IC_50_ values from 1.94 μM to 16.5 μM [55]. Six bioactive metabolites were separated from *D. phaseolorum* SKS019 derived from the mangrove plant *A. ilicifolius*, (−)-phomopsichin A (**183**), (+)-phomopsichin A (**184**), (+)-phomopsichin B (**185**), (−)-phomopsichin B (**181**), and the new diaporchromanones C (**186**) and D (**187**). These metabolites showed moderate inhibition of osteoclastogenesis by inhibiting RANKL-induced NF-_K_B activation [56]. The fungus *Diaporthe* sp. SCSIO 41011, derived from the mangrove plant *R*. *stylosa*, yielded two metabolites, *epi*-isochromophilone II (**172**) and isochromophilone D (**188**). Compound **172** displayed cytotoxicity against ACHN, OS-RC-2, and 786-cells with IC_50_ values of between 3.0 μM and 4.4 µM, and **188** had an IC_50_ of 8.9 µM against 786-O cancer cells [57]. Compound **167** showed inhibitory activity against human tumor cell lines KB and Raji with IC_50_ values of 6.25 μg/mL and 5.51 µg/mL, respectively [51]. Diaporisoindole A (**189**) and tenellone C (**190**) were obtained from *Diaporthe* sp. SYSU-HQ3 on the mangrove plant *E. agallocha* and displayed inhibitory activity on *M. tuberculosis* protein tyrosine phosphatase B (MptpB) (IC_50_ values = 4.2 μM and 5.2 µM, respectively) [58]. Eight new compounds, diaporindenes A−D (**191−194**), isoprenylisobenzofuran A (**195**), diaporisoindoles D and E (**196** and **197**), and tenellone D (**198**), were isolated from the endophytic fungus *Diaporthe* sp. SYSU-HQ3 derived from the branches of *Excoecaria agallocha.* All metabolites displayed significant anti-inflammatory activity [59]. Cordysinin A (**199**) was derived from the endophytic fungus *D*. *arecae* on *Kandelia obovate*. It displayed antiangiogenic activity against human endothelial progenitor cells (EPCs) with an IC_50_ value of 15.1 ± 0.2 μg/mL [60]. The metabolites 5-deoxybostrycoidin (**200**) and fusaristatin A (**201**) were obtained from *D*. *phaseolorum* SKS019 on the mangrove plant *A*. *ilicifolius*. Compound **200** showed cytotoxic activity against MDA-MB-435 and NCI-H460 with IC_50_ values of 5.32 μM and 6.57 μM, respectively, and the IC_50_ value of **201** on MDA-MB-435 was 8.15 μM [61]. Phomopsin F (**202**) was isolated from *D. toxica* and showed cytotoxic activity against HepG2 cells [62]. Two novel metabolites, longidiacid A (**203**) and longichalasin B (**204**), were isolated from the deep-sea-derived fungus *Diaporthe longicolla* FS429. These compounds were shown to inhibit 35.4% and 53.3% of the enzyme activity of the *Mycobacterium tuberculosis* protein tyrosine phosphatase B (MptpB), respectively, at a concentration of 50 µM [63]. The new diaporpenoid A (**205**) and the new diaporpyrone A (**206**) were isolated from a MeOH extract obtained from cultures of the endophytic mangrove fungus *Diaporthe* sp. QYM12. Compounds **205** and **206** exhibited potent anti-inflammatory activities by inhibiting the production of nitric oxide (NO) in lipopolysaccharide (LPS)-induced RAW264.7 cells with IC_50_ values of 21.5 μM and 12.5 µM, respectively [64]. Seven compounds (**168**–**174**) were isolated from the endophytic fungus *D. perseae*. Outstanding DPPH and ABTS radical scavenging activities were exhibited by all seven compounds [52]. Compounds **175**–**206** are shown in Figure 7.

### 3.3. Inactive Compounds

Secondary metabolites **207**–**221** and **124**–**134** were isolated from the mangrove-associated fungus *Diaporthe* sp. SCSIO 41011. None of these compounds reported any kind of activity [53]. Two new polyketides, phaseolorins G and H (**222** and **223**), and one new phaseolorin I (**224**), along with two known compounds (**225** and **226**), were isolated from *D. phaseolorum* FS431. None of these compounds showed any activity [54]. Two new metabolites, diaporchromanones A and B (**227** and **228**), and a known compound (**229**) were obtained from *D. phaseolorum* SKS019, but showed no activity [56]. Three chloroazaphilone derivatives (**230−232**) were obtained from the fungus *Diaporthe* sp. SCSIO 41011, along with three known analogues (**233−235**). None of these isolated compounds were reported to have any kind of activity [57]. Two inactive compounds, diaporisoindole B (**236**) and diaporisoindole C (**237**), were isolated from the endophytic fungus *Diaporthe* sp. SYSUHQ3 [58]. A new arecine (**238**) and twenty-two known diketopiperazines (**239**–**260**) were isolated from the endophytic fungus *D. arecae*, but showed no activity [60]. Six new compounds, including diaporphasines A–D (**261**–**264**) and meyeroguillines C and D (**265**–**266**), and a known meyeroguilline A (**267**) were isolated from an endophytic fungus *D. phaseolorum*. None of these compounds reported any kind of activity [61]. A chemical investigation into the fungus *D. longicolla* FS429 led to the isolation of six metabolites, the novel longidiacid B (**268**), two new polyketides (**269**–**270**), a new cytochalasin analogue longichalasins A (**272**), and two known compounds (**271** and **273**). None of them showed activity [63]. Four inactive compounds, including the new diaporpenoids B and C (**274** and **275**), and the known diaporpyrones B and C (**160** and **161**), were isolated from the mangrove endophytic fungus *Diaporthe* sp. QYM12 [64]. The structures of compounds **207**–**275** are shown in Figure 8.

In this paper, a total of 275 secondary compounds from the genus *Diaporthe* are summarized. As can be seen in Figure 9, 153 secondary metabolites were isolated from terrestrial origins and 110 from marine origins, and 12 were common to both environments. These compounds are categorized on the basis of their activity and inactivity. Figure 10 and Figure 11, and Table 1 and Table 2 show that about half of all 275 compounds reported from terrestrial and marine origins were inactive, accounting for 74 (45%) and 80 (66%) metabolites, respectively. Moreover, the active compound ratios were 56% and 34%, respectively. The active secondary metabolites showed various types of bioactivities, mainly cytotoxic (34; 20%), antibacterial (18; 11%), antifungal (14; 9%), and miscellaneous activities (26; 15%) for those of terrestrial origin and antibacterial and antifungal (10; 8%) and miscellaneous activities (32; 26%) for those of marine origin.

## 4. Analysis of Secondary Metabolite Biosynthetic Potential

Despite the numerous compounds isolated from *Diaporthe* species, recent advances in genome sequencing and bioinformatics analysis indicate that the number of biosynthetic gene clusters (BGCs) of SMs exceeds the number of SMs identified so far [65]. To fully understand SMs’ biosynthetic potential, we used the “antibiotics and secondary metabolite analysis shell–antiSMASH” tool to predict BGCs from the genomes of *Diaporthe* species available in the NCBI database (National Center for Biotechnology Information, http://www.ncbi.nlm.nih.gov/, accessed on 1 February 2023). A total of 19 species were analyzed, and the antiSMASH 7 beta was applied using the “relaxed” detection strictness. As is shown in Figure 12, most species encoded ~90 BGCs to 110 BGCs except for *Diaporthe aspalathi* (46 BGCs) and *Diaporthe helianthi* (65 BGCs).

The BGCs were characterized as polyketide (PKSs), non-ribosomal peptides (NRPSs), terpenes, hybrid PKS-NRPSs, ribosomally synthesized and post-translationally modified peptides (RiPPs), and indole-related compounds. PKSs and NRPSs are the most abundant BGCs of all species (Figure 12). Some BGCs show high similarity with known BGCs, and their SMs are common to different species (Figure 13). A number of *Diaporthe* species were predicted to synthesize alternariol, mellein, and nectriapyrone C, which were noted for their phytotoxic and antimicrobial activities [66,67,68]. These metabolites may allow organisms to inhibit competitors that occupy the same niches and facilitate invasion when organisms are acting as phytopathogens. The BGCs of enniatin, ochratoxin A, and culmorin are present in several *Diaporthe* genomes [69,70,71]. These compounds are described as “emerging mytotoxins” and are mainly produced by the *Fusarium* species, which are wheat pathogens. This indicates that not only the *Fusarium*, but also the *Diaporthe* strains can produce contaminants in food and feed. Certain compounds with medicinal potential were also observed. Clavaric acid is an inhibitor of FPTase and may be effective as an anticancer agent in tumors [72]. FR901512 is an HMG-CoA reductase inhibitor that has the potential to lower cholesterol and fat [73].

## 5. Conclusions

This review highlights the potential of the secondary metabolites of the genus *Diaporthe*. A total of 275 secondary metabolites associated with terrestrial and marine environments have been isolated from this genus during the last twelve years. We can see in Figure 9 that of the 275 compounds reported, 153 (accounting for about 55% of the total) and 110 (about 41% of the total) were derived from terrestrial and marine origins, respectively, and 12 (about 4%) were isolated in both environments. After the comprehensive literature review, we found that active metabolites (56% and 34%, respectively) are less common than inactive metabolites (45% and 66%, respectively) in terrestrial and marine environments. Moreover, a total of 92 bioactive compounds (approximately 56%) were found in terrestrial samples, while 42 (about 34%) were found in marine samples. Current studies suggest that compounds with strong bioactivities could be used as potential drug candidates in the future, but more in-depth studies are needed to explore the mechanisms involved. This study also confirms the potential of terrestrial habitats for drug discovery and will help researchers find novel natural, potent fungal products. Genomic analyses suggested that *Diaporthe* species have great potential to produce more SMs. Therefore, future efforts should be focused on activating these silent BGCs via various methods, such as changing fermentation conditions, transcriptional regulation, using chemical elicitors, and heterologous gene expression.

## Figures and Tables

**Figure 1 jof-09-00453-f001:**
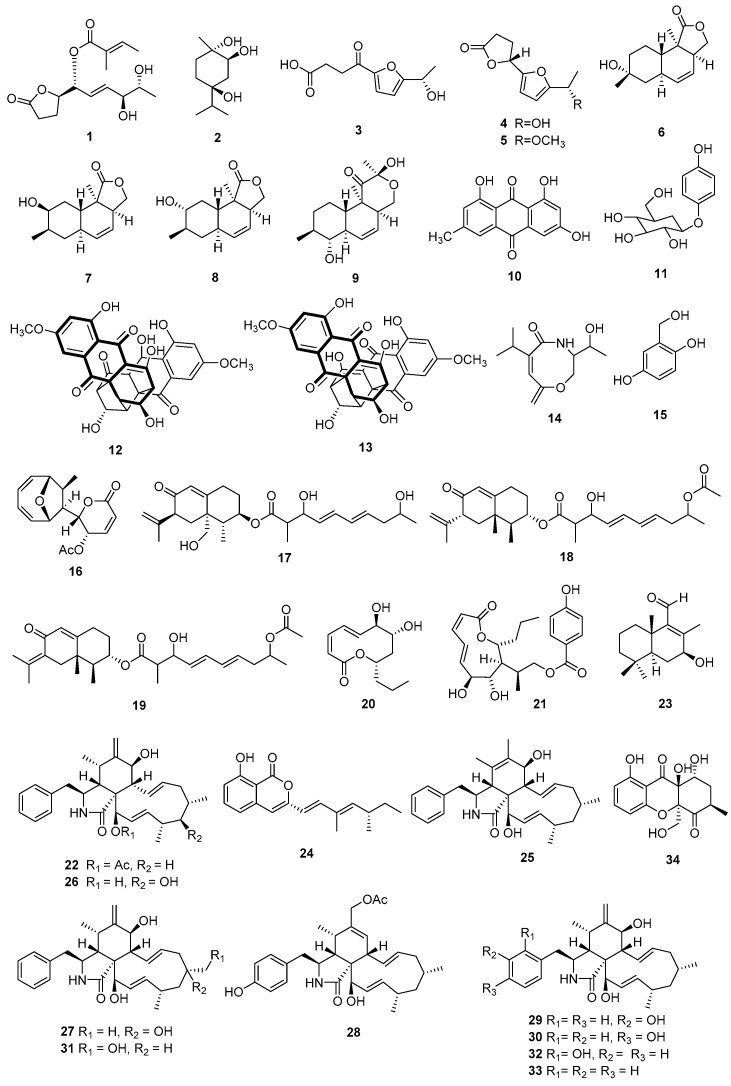
Chemical structures of compounds **1**–**34** of terrestrial origin.

**Figure 2 jof-09-00453-f002:**
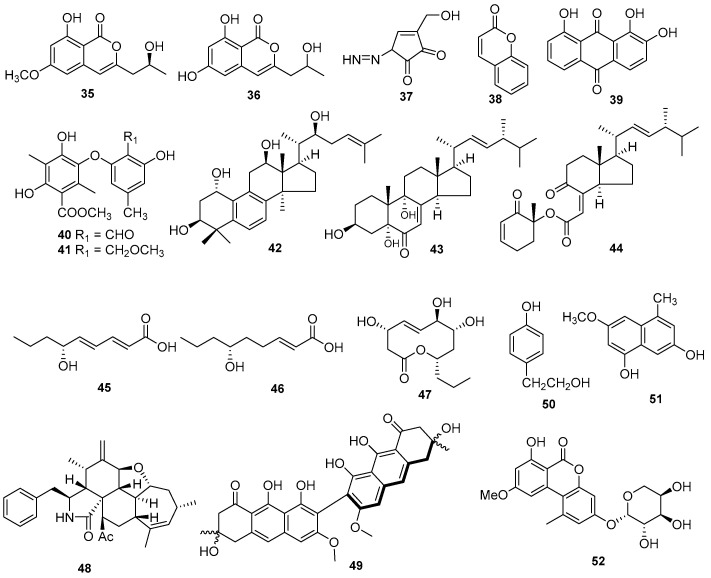
Chemical structures of compounds **35**–**52** of terrestrial origin.

**Figure 3 jof-09-00453-f003:**
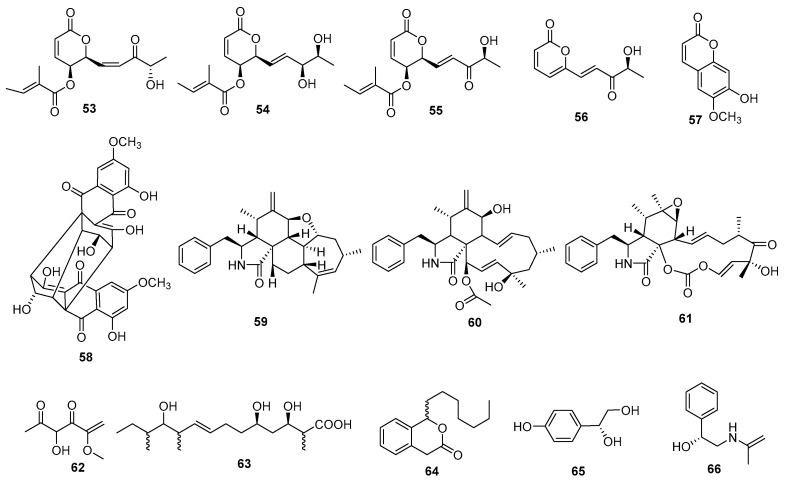
Chemical structures of compounds **53**–**66** of terrestrial origin.

**Figure 4 jof-09-00453-f004:**
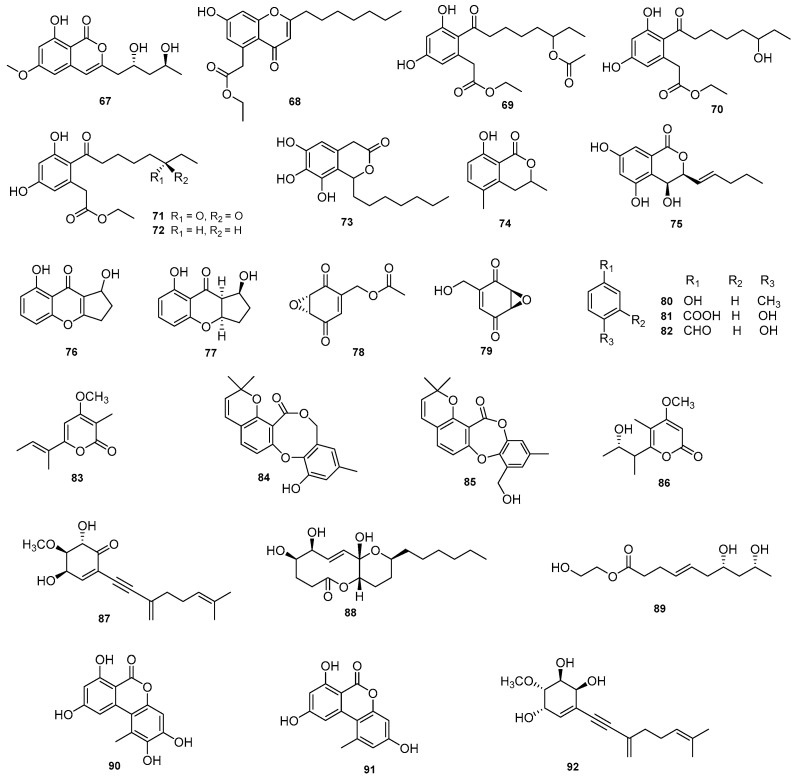
Chemical structures of compounds **67**–**92** of terrestrial origin.

**Figure 5 jof-09-00453-f005:**
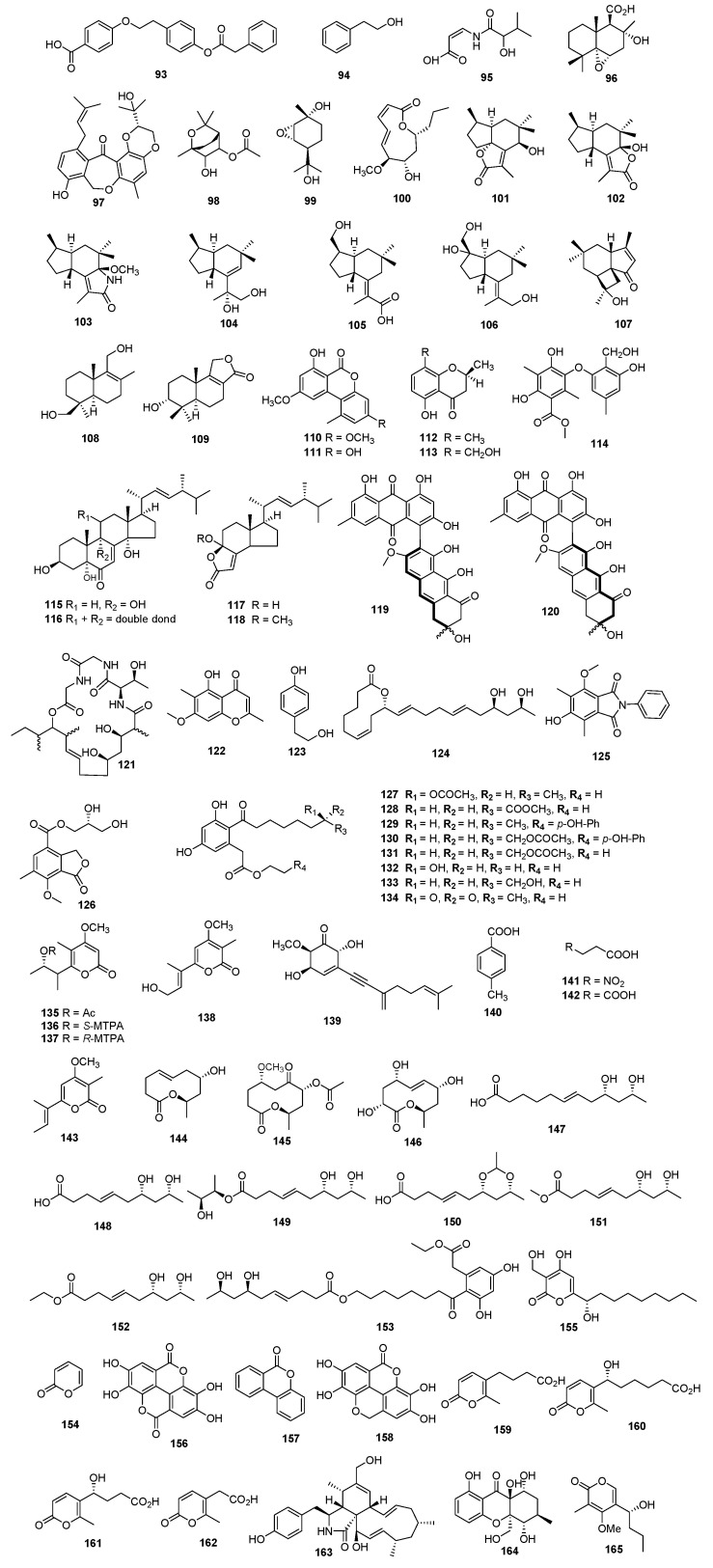
Chemical structures of compounds **93**–**165** of terrestrial origin.

**Figure 6 jof-09-00453-f006:**
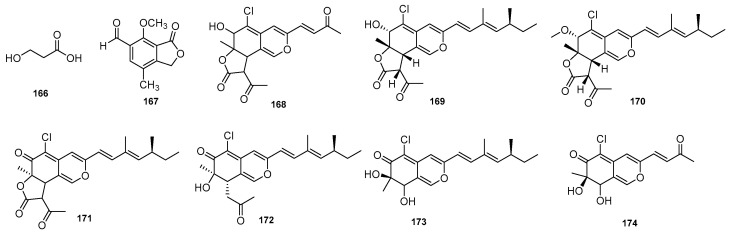
Chemical structures of compounds **166**–**174** of marine origin.

**Figure 7 jof-09-00453-f007:**
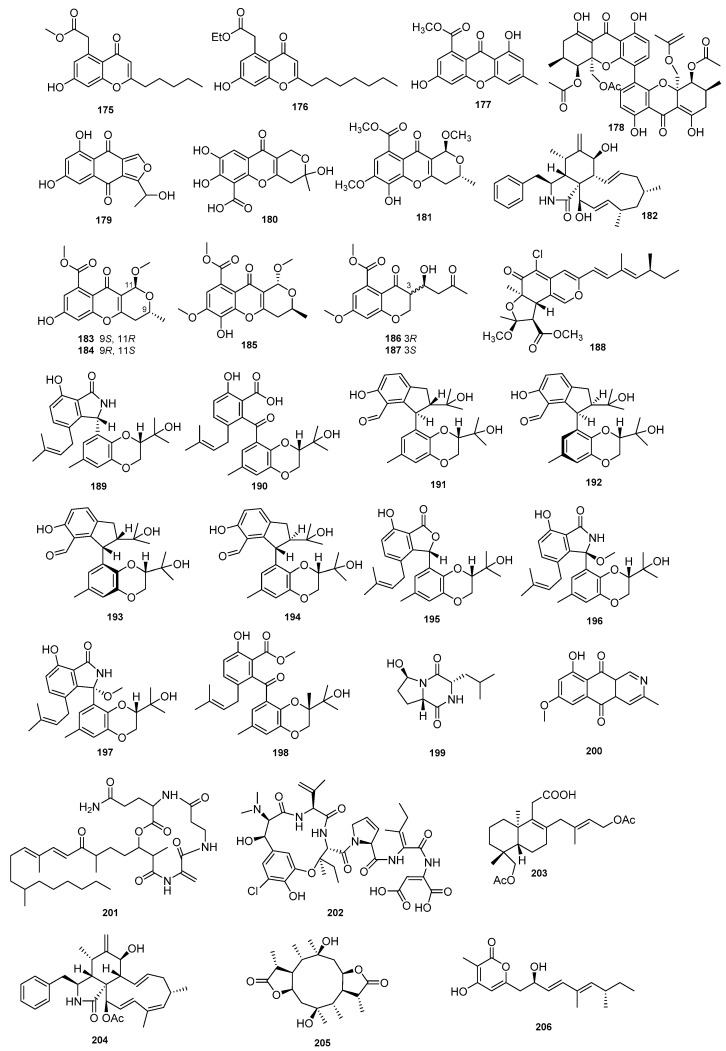
Chemical structures of compounds **175**–**206** of marine origin.

**Figure 8 jof-09-00453-f008:**
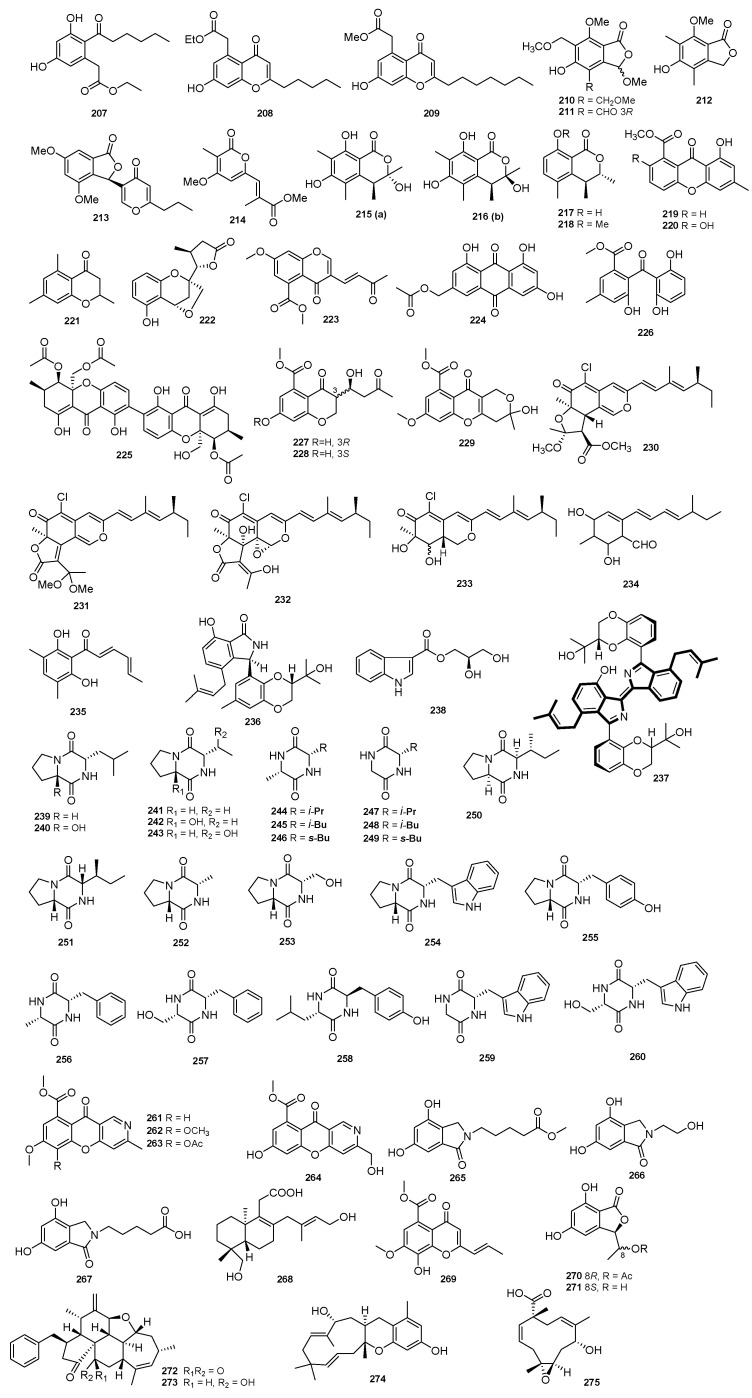
Chemical structures of compounds **207**–**275** of marine origin.

**Figure 9 jof-09-00453-f009:**
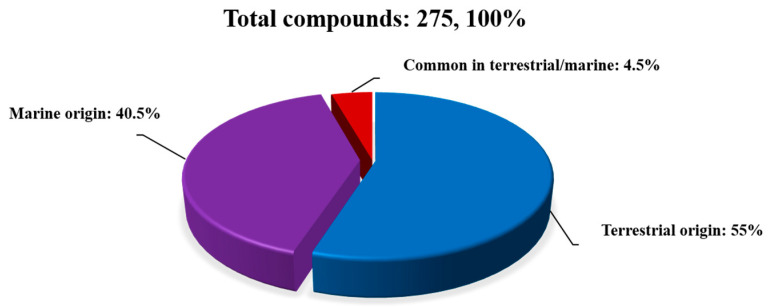
Total number of compounds isolated from genus *Diaporthe*.

**Figure 10 jof-09-00453-f010:**
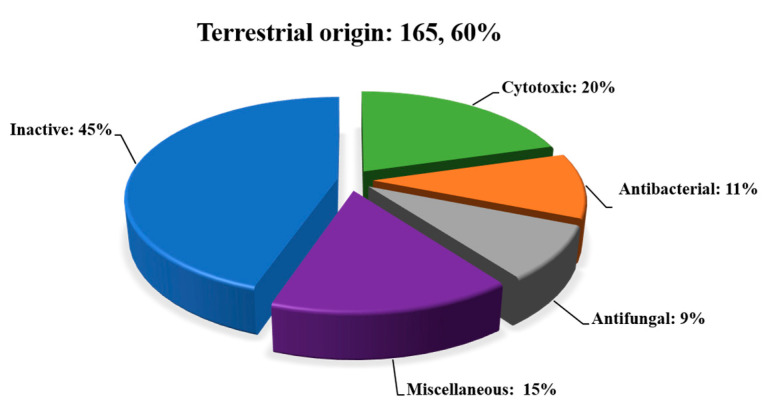
The proportion of secondary metabolites of terrestrial origin.

**Figure 11 jof-09-00453-f011:**
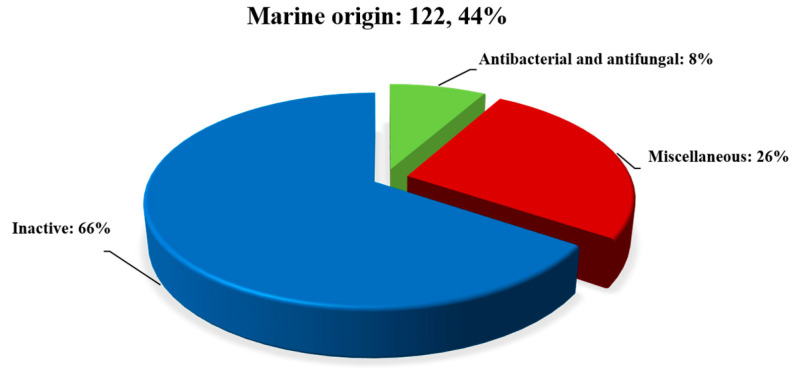
The proportion of secondary metabolites of marine origin.

**Figure 12 jof-09-00453-f012:**
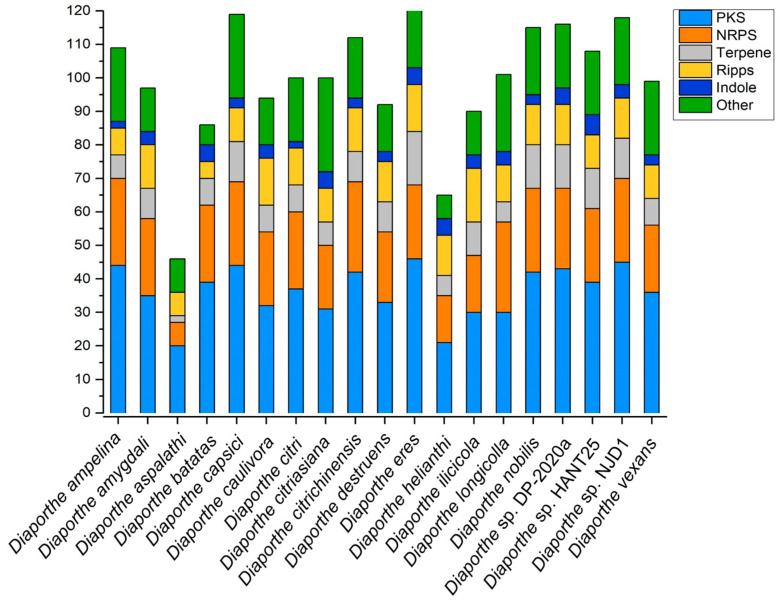
The number (*y*-axis) and type of secondary-metabolite BGCs in *Diaporthe* strains deposited in the NCBI database.

**Figure 13 jof-09-00453-f013:**
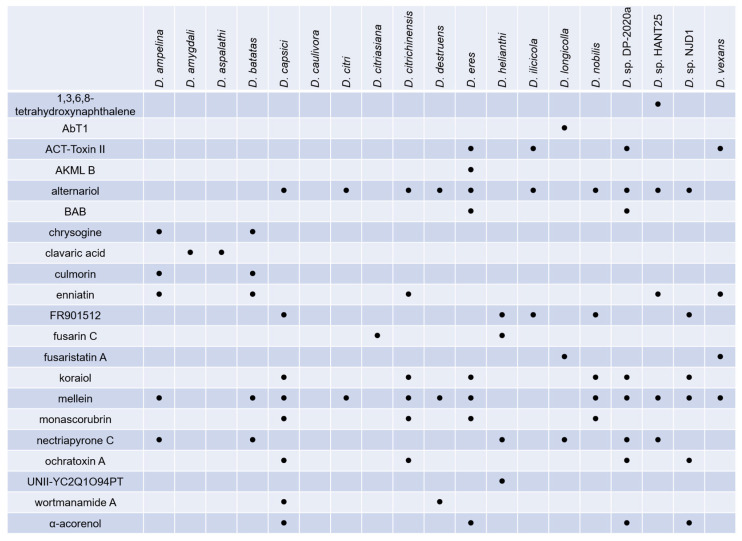
Some secondary compounds produced by species of *Diaporthe* that are 100% identical to known BGCs.

**Table 1 jof-09-00453-t001:** Secondary metabolites associated with terrestrial origin.

No.	Compound	Producing Strain	Active/Inactive	Ref.
1	(1*R*,2*E*,4*S*,5*R*)-1-[(2*R*)-5-oxotetrahydrofuran-2-yl]-4,5-dihydroxy-hex-2-en-1-yl(2*E*)-2 methylbut-2-enoate	*Diaporthe* sp. SXZ-19	Cytotoxic	[23]
2	(1*R*,2*R*,4*R*)-trihydroxy-*p*-menthane	−	Cytotoxic	−
3	butyl 5-[(1*R*)-1-hydroxyethyl]-γ-oxofuran-2-butanoate	−	Cytotoxic	−
4	3,4-dihydro-5′-[(1*R*)-1-hydroxyethyl] [2,2′-bifuran]-5(2*H*)-one	−	Cytotoxic	−
5	3,4-dihydro-5′-[(1*R*)-1-hydroxymethylethyl] [2,2′-bifuran]-5(2*H*)-one	−	Cytotoxic	−
6	Oblongolides D	−	Cytotoxic	−
7	Oblongolides H	−	Cytotoxic	−
8	Oblongolides P	−	Cytotoxic	−
9	Oblongolides V	−	Cytotoxic	−
10	Emodin	*D. lithocarpus*	Cytotoxic, Antibacterial	[15]
11	Arbutin	−	Cytotoxic	−
12	Cytoskyrin C	*Diaporthe* sp.	Cytotoxic, Activate the NF-_K_B pathway	[24]
13	(+)-epicytoskyrin	−	Cytotoxic, Activate the NF-_K_B pathway	−
14	Vochysiamide B	*D. vochysiae* LGMF1583	Cytotoxic, Antibacterial	[8]
15	2,5-dihydroxybenzyl alcohol	−	Cytotoxic	−
16	Mycoepoxydiene	*Diaporthe* sp. SNB-GSS10	Cytotoxic	[6]
17	Eremofortin F	−	Cytotoxic	−
18	Lithocarins B	*D. lithocarpus* A740	Cytotoxic	[9]
19	Lithocarins C	−	Cytotoxic	−
20	Xylarolide	*D. terebinthifolii* GG3F6	Cytotoxic, Antibacterial	[12]
21	Xylarolide A	*Diaporthe* sp.	Cytotoxic	[25]
22	18-des-hydroxy cytochalasin H	*D. phaseolorum*-92C	Cytotoxic, Antioxidant	[26]
23	Diaporol R	*Diaporthe* sp.	Cytotoxic	[27]
24	Diaporone A	*Diaporthe* sp.	Cytotoxic, Antibacterial	[28]
25	Diaporthichalasin D	*Diaporthe* sp. SC-J0138	Cytotoxic	[29]
26	Diaporthichalasin E	−	Cytotoxic	−
27	Diaporthichalasin F	−	Cytotoxic	−
28	Diaporthichalasin H	−	Cytotoxic	−
29	Diaporthichalasin A	−	Cytotoxic	−
30	Diaporthichalasin B	−	Cytotoxic	−
31	Diaporthichalasin C	−	Cytotoxic	−
32	Phomopsichalasin G	−	Cytotoxic	−
33	21-O-deacetyl-L-696,474	−	Cytotoxic	−
34	Phomopthane A	*D. unshiuensis* YSP3	Cytotoxic	[30]
35	(10*S*)-diaporthin	*D. terebinthifolii* LGMF907	Antibacterial	[31]
36	Orthosporin	−	Antibacterial	−
37	Kongiidiazadione	*D. kongii*	Antibacterial	[32]
38	Coumarin	*D. lithocarpus*	Antibacterial	[15]
39	1,2,8-trihydroxyanthraquinone	−	Antibacterial	−
40	Phomosines A	*Diaporthe* sp. F2934	Antibacterial	[11]
41	Phomosines C	−	Antibacterial	−
42	19-nor-lanosta-5(10),6,8,24-tetraene- 1*α*,3*β*,12*β*,22*S*-tetraol	*Diaporthe* sp. LG23	Antibacterial	[14]
43	3*b*,5*a*,9*a*-trihydroxy-(22E,24R)-ergosta-7,22-dien-6-one	−	Antibacterial	−
44	Chaxine C	−	Antibacterial	−
45	Diapolic acid A	*D*. *terebinthifolii* GG3F6	Antibacterial	[12]
46	Diapolic acid B	−	Antibacterial	−
47	Phomolide G	−	Antibacterial	−
48	21-acetoxycytochalasins J_3_	*Diaporthe* sp. GDG-118	Antibacterial	[33]
49	Flavomannin-6,60-di-O-methyl ether	*D. melonis*	Antibacterial	[34]
50	Tyrosol	*D. helianthi, D. eres*	Antibacterial, Phytotoxic	[35,39,42]
51	3-methoxy-5-methylnaphthalene-1, 7-diol	*Diaporthe* sp.	Antibacterial	[36]
52	Alternariol methyl ether-12-O-α-D-arabinoside	*D. unshiuensis* YSP3	Antibacterial	[30]
53	Phomopsolide A	*D. maritima*	Antifungal	[5]
54	Phomopsolide B	−	Antifungal	−
55	Phomopsolide C	−	Antifungal	−
56	(*S*,*E*)-6-(4-hydroxy-3-oxopent-1-en-1-yl)-2H-pyran-2-one	−	Antifungal	−
57	7-hydroxy-6-metoxycoumarin	*D. lithocarpus*	Antifungal	[15]
58	(+)-2,20-epicytoskyrin A	*Diaporthe* sp. GNBP-10	Antifungal	[37]
59	7-acetoxycytochalasin H	*Diaporthe* sp. GDG-118	Antifungal	[32]
60	Cytochalasin H	−	Antifungal	−
61	Cytochalasin E	−	Antifungal	−
62	3-hydroxy-5-methoxyhex-5-ene-2,4-dione	*Diaporthe* sp. ED2	Antifungal	[16]
6	Eucalyptacid A	*D. eucalyptorum* KY-9	Antifungal	[13]
64	Cytosporone C	−	Antifungal	−
65	1-(4-hydroxyphenyl) ethane-1,2-diol	−	Antifungal	−
66	(2-hydroxy-2-phenylethyl) acetamide	−	Antifungal	−
67	Mucorisocoumarin A	*D. pseudomangiferaea*	Antifibrosis	[38]
68	Pestalotiopsone B	−	Antifibrosis	−
69	Acetoxydothiorelone B	−	Antifibrosis	−
70	Dothiorelone B	−	Antifibrosis	−
71	Dothiorelone L	−	Antifibrosis	−
72	Dothiorelone G	−	Antifibrosis	−
73	Cytosporone D	−	Antifibrosis, Antioxidant, Antidiabetic	−
74	3,4-dihydro-8-hydroxy-3,5- dimethylisocoumarin	*D. eres*	Phytotoxic	[39]
75	Diportharine A	*Diaporthe* sp.	Antioxidant	[25]
76	Diaportheone A	*Diaporthe* sp. P133	Antitubercular	[40]
77	Diaportheone B	−	Antitubercular	−
78	Phyllostine acetate	*D. miriciae*	Antifeedant	[41]
79	Phyllostine	−	Antifeedant	−
80	P-cresol	*D. eres*	Phytotoxic	[42]
81	4-hydroxybenzoic acid	−	Phytotoxic	−
82	4-hydroxybenzaldehyde	−	Phytotoxic	−
83	Nectriapyrone	−	Phytotoxic	−
84	Diaporthol A	*Diaporthe* sp. ECN-137	Antimigratory	[43]
85	Diaporthol B	−	Antimigratory	−
86	Gulypyrone A	*D*. *gulyae*	Phytotoxic	[44]
87	Phomentrioloxin B	−	Phytotoxic	−
88	Phomolide C	*Diaporthe* sp.	Antiproliferation effect	[45]
89	Diaporthsin E	*Diaporthe* sp. JC-J7	Antihyperlipidemic	[46]
90	2-hydroxy-alternariol	*Diaporthe* sp. CB10100	Reduced NO production	[47]
91	Alternariol	−	Reduced NO production	−
92	Phomentrioloxin	*Phomopsis* sp.	Phytotoxic	[48]
93	Diaporthindoic acid	*D. lithocarpus*	Inactive	[15]
94	2-phenylethanol	−	Inactive	−
95	Vochysiamides A	*D. vochysiae* LGMF1583	Inactive	[8]
96	Altiloxin A	*D. pseudomangiferae*	Inactive	[6]
97	Tenllone I	*D. lithocarpus* A740	Inactive	[9]
98	Lithocarin D	−	Inactive	−
99	Phomopene	−	Inactive	−
100	Xylarolide B	*Diaporthe* sp.	Inactive	[25]
101	Diaporol J	*Diaporthe* sp.	Inactive	[27]
102	Diaporol K	−	Inactive	−
103	Diaporol L	−	Inactive	−
104	Diaporol M	−	Inactive	−
105	Diaporol N	−	Inactive	−
106	Diaporol O	−	Inactive	−
107	Diaporol P	−	Inactive	−
108	Diaporol Q	−	Inactive	−
109	Diaporol S	−	Inactive	−
110	Alternariol 4,10-dimethyl ether	*Diaporthe* sp.	Inactive	[28]
111	Alternariol 4-methyl ether	−	Inactive	−
112	4H-1-benzopyra-4-one-2,3-dihydro-5-hydroxy-2,8-dimetyl	*Diaporthe* sp. F2934	Inactive	[11]
113	4H-1-benzopyran-4-one-2,3-dihydro-5-hydroxy-8-(hydroxy-lmethyl)-2-methyl	−	Inactive	−
114	Phomosine D	−	Inactive	−
115	3*β*,5*α*,9*α*,14*α*-tetrahydroxy-(22*E*,24*R*)-ergosta-7,22-dien 6-one	*Diaporthe* sp. LG23	Inactive	[14]
116	(22*E*,24*R*)-ergosta-7,9(11),22-triene-3*β*,5*α*,6*α*-triol	−	Inactive	−
117	Demethylincisterol A3	−	Inactive	−
118	Volemolide	−	Inactive	−
119	Diaporthemin A	*D. melonis*	Inactive	[34]
120	Diaporthemin B	−	Inactive	−
121	Eucalactam B	*D. eucalyptorum* KY-9	Inactive	[13]
122	Eugenitol	−	Inactive	−
123	4-hydroxyphenethyl alcohol	−	Inactive	−
124	(9*S*, 17*R*, 19*S*, 6*Z*, 10*E*, 14*E*)-Diaporlactone A	*D. pseudomangiferaea*	Inactive	[38]
125	5-hydroxy-7-methoxy-4,6-dimethyl-2-phenylisoindoline-1,3-dione	−	Inactive	−
126	(13*R*)-Diaporphthalide	−	Inactive	−
127	(15*S*)-Acetoxydothiorelone A	−	Inactive	−
128	Dothiorelone K	−	Inactive	−
129	Dothiorelone M	−	Inactive	−
130	Dothiorelone N	−	Inactive	−
131	16-acetoxydothiorelone C	−	Inactive	−
132	Dothiorelone A	−	Inactive	−
133	Dothiorelone C	−	Inactive	−
134	Dothiorelone I	−	Inactive	−
135	9-*O*-acetyl derivative	*D. gulyae*	Inactive	[44]
136	9-*O*-*S*-MTPA ester	−	Inactive	−
137	9-*O*-*R*-MTPA ester	−	Inactive	−
138	Gulypyrone B	−	Inactive	−
139	Phomentrioloxin C	−	Inactive	−
140	4-methylbenzoic acid	−	Inactive	−
141	3-nitropropionic acid	−	Inactive	−
142	Succinic acid	−	Inactive	−
143	Nectryapyrone	−	Inactive	−
144	Diaporthsin A	*Diaporthe* sp. JC-J7	Inactive	[46]
145	Diaporthsin F	−	Inactive	−
146	Diaporthsin H	−	Inactive	−
147	Diaporthsin C	−	Inactive	−
148	Diaporthsin B	−	Inactive	−
149	Diaporthsin D	−	Inactive	−
150	Diaporthsin G	−	Inactive	−
151	Diaporthsin I	−	Inactive	−
152	Diaporthsin J	−	Inactive	−
153	Diaporthsin K	−	Inactive	−
154	α-Pyrone	*Diaporthe* sp. CB10100	Inactive	[47]
155	Dothideopyrone F	−	Inactive	−
156	Ellagic acid	−	Inactive	−
157	Dibenzo-α-pyrone	−	Inactive	−
158	Ellagic acid B	−	Inactive	−
159	Diaporpyrone A	−	Inactive	−
160	Diaporpyrone B	−	Inactive	−
161	Diaporpyrone C	−	Inactive	−
162	Diaporpyrone D	−	Inactive	−
163	Diaporthichalasin G	*Diaporthe* sp. SC-J0138	Inactive	[29]
164	Phomopthane B	*D. unshiuensis* YSP3	Inactive	[30]
165	Phomopyrone B	*−*	Inactive	[30]

**Table 2 jof-09-00453-t002:** Secondary metabolites associated with marine origin.

No.	Compound	Producing Strain	Active/Inactive	Ref.
60	Cytochalasin H	*Diaporthe amygdali* SgKB4, *Diaporthe* sp. GZU-1021	Antibacterial, Anti-NO production	[49,55]
166	3-hydroxypropionic acid	*D. phaseolorum*	Antibacterial	[50]
167	Diaporthelactone	*Diaporthe* sp.	Antifungal, Cytotoxic	[51]
168	Isochromophilone G	*D. perseae*	Antibacterial, Anti-inflammatory	[52]
169	Isochromophilone A	−	Antibacterial, Anti-inflammatory	−
170	Isochromophilone B	−	Antibacterial, Anti-inflammatory	−
171	5-chloroisorotiorin	−	Antibacterial, Anti-inflammatory	−
172	*epi*-isochromophilone II	*D. perseae*, *Diaporthe* sp. SCSIO 41011	Antibacterial, Cytotoxic, Anti-inflammatory	[52,57]
173	Isochromophilone III	*D. perseae*	Antibacterial, Anti-inflammatory	[52]
174	Penicilazaphilone D	*D. perseae*	Antibacterial, Anti-inflammatory	[52]
175	Pestalotiopsones F	*Diaporthe* sp. SCSIO 41011	Anti-IAV	[53]
176	Pestalotiopsones B	−	Anti-IAV	−
177	3,8-dihydroxy-6-methyl-9-oxo-9Hxanthene- 1-carboxylate	−	Anti-IAV	−
178	Phomoxanthone A	*D. phaseolorum* FS431	Cytotoxic	[54]
179	Biatriosporin N	*Diaporthe* sp. GZU-1021	Anti-NO production	[55]
180	Penialidin A	−	Anti-NO production	−
181	(−)-phomopsichin B	*Diaporthe* sp. GZU-1021, *D. phaseolorum* SKS019	Anti-NO production, Antiosteoclastogenesis	[55,56]
182	21-O-deacetyl-L-696,474	*Diaporthe* sp. GZU-1021	Anti-NO production	[55]
183	(−)-phomopsichin A	*D. phaseolorum* SKS019	Antiosteoclastogenesis	[56]
184	(+)-phomopsichin A	−	Antiosteoclastogenesis	−
185	(+)-phomopsichin B	−	Antiosteoclastogenesis	−
186	Diaporchromanone C	−	Antiosteoclastogenesis	−
187	Diaporchromanone D	−	Antiosteoclastogenesis	−
188	Isochromophilone D	*Diaporthe* sp. SCSIO 41011	Cytotoxic	[57]
189	Diaporisoindole A	*Diaporthe* sp. SYSU-HQ3	Cytotoxic	[58]
190	Tenellone C	−	Cytotoxic	−
191	Diaporindene A	*Diaporthe* sp. SYSU-HQ3	Anti-inflammatory	[59]
192	Diaporindene B	−	Anti-inflammatory	−
193	Diaporindene C	−	Anti-inflammatory	−
194	Diaporindene D	−	Anti-inflammatory	−
195	Isoprenylisobenzofuran A	−	Anti-inflammatory	−
196	Diaporisoindole D	−	Anti-inflammatory	−
197	Diaporisoindole E	−	Anti-inflammatory	−
198	Tenellone D	−	Anti-inflammatory	−
199	Cordysinin A	*D*. *arecae*	Antiangiogenic	[60]
200	5-deoxybostrycoidin	*D. phaseolorum* SKS019	Cytotoxic	[61]
201	Fusaristatin A	−	Cytotoxic	−
202	Phomopsin F	*D. toxica*	Cytotoxic	[62]
203	Longidiacid A	*Diaporthe longicolla* FS429	Enzymatic activity	[63]
204	Longichalasin B	−	Enzymatic activity	−
205	Diaporpenoid A	*Diaporthe* sp. QYM12	Anti-inflammatory	[64]
206	Diaporpyrone A	−	Anti-inflammatory	−
207	Secocurvularin	*Diaporthe* sp. SCSIO 41011	Inactive	[53]
208	Pestalotiopsone H	−	Inactive	−
209	Pestalotiopsone A	−	Inactive	−
210	(±)-microsphaerophthalide H	−	Inactive	−
211	Microsphaerophthalide I	−	Inactive	−
212	5-hydroxy-7-methoxy-4,6-dimethylphthalide	−	Inactive	−
213	Dihydrovermistatin	−	Inactive	−
214	Methyl convolvulopyrone	−	Inactive	−
215	Sclerotinin A (a)	−	Inactive	−
216	Sclerotinin A (b)	−	Inactive	−
217	3,5-dimethyl-8-hydroxy-3,4-dihydroisocoumarin	−	Inactive	−
218	3,5-dimethyl-8-methoxy-3,4-dihydroisocoumarin	−	Inactive	−
219	methyl 8-hydroxy-6-methyl-9-oxo-9Hxanthene-1-carboxylate	−	Inactive	−
220	Pinselin	−	Inactive	−
221	7-hydroxy-2,5-dimethylchromone	−	Inactive	−
222	Phaseolorin G	*D. phaseolorum* FS431	Inactive	[54]
223	Phaseolorin H	−	Inactive	−
224	Phaseolorin I	−	Inactive	−
225	Dicerandrol B	−	Inactive	−
226	2,20,60-trihydroxy-4-methyl-6-methoxy-acyl-diphenylmethanone	−	Inactive	−
227	Diaporchromanone A	*D. phaseolorum* SKS019	Inactive	[56]
228	Diaporchromanone B	−	Inactive	−
229	(±)-diaporchromone A	−	Inactive	−
230	Isochromophilone C	*Diaporthe* sp. SCSIO 41011	Inactive	[57]
231	Isochromophilone E	−	Inactive	−
232	Isochromophilone F	−	Inactive	−
233	*epi*-isochromophilone III	−	Inactive	−
234	6-((1*E*,3*E*)-3,5-dimethylhepta-1,3-dien-1-yl)-2,4-dihydroxy-3-methylbenzaldehyde	−	Inactive	−
235	(2*E*,4*E*)-1-(2,6-dihydroxy-3,5-dimethylphenyl)hexa-2,4-dien-1-one)	−	Inactive	−
236	Diaporisoindole B	*Diaporthe* sp. SYSUHQ3	Inactive	[58]
237	Diaporisoindole C	−	Inactive	−
238	Arecine	*D. arecae*	Inactive	[60]
239	Cyclo(_L_-Thr-_L_-Pro)	−	Inactive	−
240	Cyclo(6-hydroxy-Pro-_L_-Leu)	−	Inactive	−
241	Cyclo(_L_-Val-_L_- Pro)	−	Inactive	−
242	Bacillusamide B	−	Inactive	−
243	Cyclo(_L_-Leu-_L_-Pro)	−	Inactive	−
244	Cyclo(_L_-Val-_L_-Ala)	−	Inactive	−
245	Cyclo(_L_-Leu-_L_-Ala)	−	Inactive	−
246	Cyclo(_L_-Ile-_L_-Ala)	−	Inactive	−
247	Cyclo(Gly-_L_-Val)	−	Inactive	−
248	Cyclo(Gly-_L_-Leu)	−	Inactive	−
249	Cyclo(Gly-_L_-Ile)	−	Inactive	−
250	Cyclo(_L_-Ile-_D_-Pro)	−	Inactive	−
251	Staphyloamide A	−	Inactive	−
252	Cyclo(_L_-Ala-_L_-Pro)	−	Inactive	−
253	Cyclo(_L_-Ser-_L_-Pro)	−	Inactive	−
254	Cyclo(_L_-Trp-_L_-Pro)	−	Inactive	−
255	Cyclo(_L_-Tyr-_L_-Pro)	−	Inactive	−
256	Cyclo(_L_-Phe-_L_-Ala)	−	Inactive	−
257	Cyclo(_L_-Ser-_L_-Phe)	−	Inactive	−
258	Cyclo(_D_-Tyr-_L_-Leu)	−	Inactive	−
259	Cyclo(Gly-_L_-Trp)	−	Inactive	−
260	Cyclo(_L_-Trp-_L_-Ser)	−	Inactive	−
261	Diaporphasine A	*D. phaseolorum*	Inactive	[61]
262	Diaporphasine B	−	Inactive	−
263	Diaporphasine C	−	Inactive	−
264	Diaporphasine D	−	Inactive	−
265	Meyeroguilline C	−	Inactive	−
266	Meyeroguilline D	−	Inactive	−
267	Meyeroguilline A	−	Inactive	−
268	Longidiacid B	*D. longicolla* FS429	Inactive	[63]
269	Longichromone A	−	Inactive	−
270	Longiphthalidin A	−	Inactive	−
271	Acetophthalidin	−	Inactive	−
272	Longichalasin A	−	Inactive	−
273	Cytochalasin J3	−	Inactive	−
274	Diaporpenoid B	*Diaporthe* sp. QYM12	Inactive	[64]
275	Diaporpenoid C	−	Inactive	−

## Data Availability

Not applicable.

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
