# Peer review of "Potential of Secondary Metabolites of Diaporthe Species Associated with Terrestrial and Marine Origins"

_jof, 2023, doi:10.3390/jof9040453_

Round 1

Reviewer 1 Report

This paper provides a comprehensive overview on the structures and bioactivities of genus Diaporthe associated with terrestrial and marine origins. The whole manuscript is complete, whereas the following suggestions can be considered.

1.       Many expressions are colloquial and the English language in this manuscript should be improved.

2.       The tense in different sentences should be consistent.

3.       Some predicate and non-predicate verbs are used incorrectly.

(1) 2.1. Cytotoxic metabolites, line 18, “was” should be changed into “were”.

(2) 2.2. Antibacterial metabolites, line 26, “isolated from” should be changed into “ were isolated from”.

(3) 3.1. Antibacterial and antifungal metabolites, line 9, “exhibited” should be changed into “exhibiting”.

In this manuscript, there are many similar mistakes. Please check it carefully.

4.       Some words in this manuscript are not appropriate.

(1) Page 2, line 17: “a difficult approach” => “difficult”

(2) Page 2, line 19: “separated” => “classified”

(3) 2.2. Antibacterial metabolites, line 2: “extracted” => “isolated”

In this manuscript, there are many similar mistakes. Please check it carefully.

5.       The structures of the compounds in figures should be arranged more neatly and aesthetically.

6.       Some structures of the compounds aren’t standard. For example, the bond angle in some structures is not 120° and the sizes of font are inconsistent. Please check all the structures carefully.

7.       “2.1.Antifungal secondary metabolites” => “2.3. Antifungal secondary metabolites” There are some similar mistakes. Please check it carefully.

8.       Introduction, line 1: “Diaporthe”=>“Diaporthe

9.       Introduction, line 6: “It consists of about 800 species, ignoring the >950 species assigned to its asexual state (Phomopsis).” This sentence presents the language disorder.

10.    Page 2, line 9: “host type”=>“host types”

11.    Page 2, line 9: “ Fungal endophytes have been found to occupy several niches in their natural ecosystem that determine their functional aspects, including tolerance to abiotic and biotic stressors to increase their resistance, help plants adapt to new habitats, and protect them from various pests and pathogens. In return, they benefitted from host plants in several ways, including providing nutrients, protection from desiccation, spatial structure, and passing on reproductive fungal propagules to the next generation of hosts in case of vertical transmission.” This sentence presents the obscure meaning and language disorder.

12.    2.2. Antibacterial metabolites, line 20, “22E,24R”=>“22E,24R

13.     2.1. Antifungal secondary metabolites, line 8, “bis-anthraquinone” => “bis-anthraquinone”

14.     2.1. Antifungal secondary metabolites, line 8, “(+)-2,20-epi-cytoskyrin A” => “(+)-2,20-epi-cytoskyrin A”

15.    Page 3, line 7: “inhibited cytotoxic activity” => “exhibited cytotoxic activity”. There are many similar mistakes. Please check it carefully.

16.    Page 8, line 25: “showen” => “showed”

17.    Page 10, line 7: “Chemical investigation of the endophytic fungus D. melonis, yielded two new compounds, diaporthemins A and B.” This sentence presents the language disorder.

18.    Page 10, line 7: “three known” => “two known”

19.    3.1. Antibacterial and antifungal metabolites, line 11: “and along with” =>“along with”

20.    3.1.Miscellaneous activities, line 5: “Phomoxanthone A with novel carbon skeleton was isolated from the fungus D. phaseolorum FS431 from deep-sea sediment of the Indian Ocean revealed good cytotoxicity against MCF-7, HepG-2, and A549 with IC50 values of 2.60, 2.55, and 4.64 µM, respectively.” This sentence presents the language disorder.

21.    Page 18, line 3: “IC50 = values 4.2 and 5.2 µM” => “IC50 values = 4.2 and 5.2 µM”

22.    Page 18, line 20: “a new diaporpyrones A” => “a new diaporpyrone A”

23.    Page 21, line 6: “The chemical investigation fungus” => “The chemical investigation of fungus”

24.    Page 21, line 8: “analogues” => “analogue”

25.    Page 26, line 2: “Figure 9” => “Figure 9”. There are some similar mistakes. Please check it carefully.

26.    Page 26, line 9: “various type” => “various types”

27.    Page 28: Figure 12, “Diaporthe” => “Diaporthe

28.    Page 29, line 3: “Here, we can see from figure 9 that among all of the reported compounds 271” This sentence presents the language disorder.

29.    Page 29, line 3: “we found more bioactive compounds 89 (about 55%) from terrestrial origin” This sentence presents the language disorder.

Page 29, line 10: “According to present studies, compounds with potent bioactivities may serve as potential drug candidates in the future, but in-depth studies needed to explore the mechanisms involved.” This sentence presents the language disorder.

Reviewer 2 Report

research is well organized however some sections are overlapping (especially in the introduction).

in the discussion, authors must cite some more articles and elaborate your concluding remarks.

Reviewer 3 Report

The manuscript entitled, “Secondary metabolites potential of genus Diaporthe associated with terrestrial and marine origins” is interesting. Overall, the review highlighting the potential of SMs belonging to genus Diaporthe is nicely presented. However, there are a few comments that need to be addressed.

 Major comments:

1.       In sections 2.1 and 2.2, all the information has been given in one large paragraph. It is very difficult to precisely follow the studies. This sentence should be divided into several small relevant paragraphs. Please follow the same trend for other subheadings.

2.       Also, the sections 2.1 and 2.2 are repeated. For example, section 2.1. covers two subheadings, “Cytotoxic metabolites and Antifungal secondary metabolites”.  

 Minor comments:

1.       Diaporthe should be “italicized” when appear first in the “Introduction” section. 
